# Exploring the Impact of Copper Oxide Substitution on Structure, Morphology, Bioactivity, and Electrical Properties of 45S5 Bioglass^®^

**DOI:** 10.3390/biomimetics9040213

**Published:** 2024-04-02

**Authors:** Imen Hammami, Manuel Pedro Fernandes Graça, Sílvia Rodrigues Gavinho, Suresh Kumar Jakka, João Paulo Borges, Jorge Carvalho Silva, Luís Cadillon Costa

**Affiliations:** 1I3N and Physics Department, University of Aveiro, 3810-193 Aveiro, Portugal; imenhammami@ua.pt (I.H.); mpfg@ua.pt (M.P.F.G.); silviagavinho@ua.pt (S.R.G.); suresh@ua.pt (S.K.J.); 2CENIMAT-I3N and Materials Science Department, NOVA School of Science and Technology, Campus de Caparica, 2829-516 Caparica, Portugal; jpb@fct.unl.pt; 3CENIMAT-I3N and Physics Department, NOVA School of Science and Technology, Campus de Caparica, 2829-516 Caparica, Portugal; jcs@fct.unl.pt

**Keywords:** bioglass^®^, biomaterial, implant coatings, osseointegration, electrical properties

## Abstract

In recent decades, the requirements for implantable medical devices have increased, but the risks of implant rejection still exist. These issues are primarily associated with poor osseointegration, leading to biofilm formation on the implant surface. This study focuses on addressing these issues by developing a biomaterial for implant coatings. 45S5 bioglass^®^ has been widely used in tissue engineering due to its ability to form a hydroxyapatite layer, ensuring a strong bond between the hard tissue and the bioglass. In this context, 45S5 bioglasses^®^, modified by the incorporation of different amounts of copper oxide, from 0 to 8 mol%, were synthesized by the melt–quenching technique. The incorporation of Cu ions did not show a significant change in the glass structure. Since the bioglass exhibited the capacity for being polarized, thereby promoting the osseointegration effectiveness, the electrical properties of the prepared samples were studied using the impedance spectroscopy method, in the frequency range of 10^2^–10^6^ Hz and temperature range of 200–400 K. The effects of CuO on charge transport mobility were investigated. Additionally, the bioactivity of the modified bioglasses was evaluated through immersion tests in simulated body fluid. The results revealed the initiation of a Ca–P-rich layer formation on the surface within 24 h, indicating the potential of the bioglasses to enhance the bone regeneration process.

## 1. Introduction

Nowadays, the scientific field of biomaterials has gained great attention. Researchers are focused on the development of biomaterials compatible with the human body to preserve the physical integrity and comfort of people with functional impairments or victims of injuries [1,2,3]. Historically, several materials, such as metallic components, ceramics, polymers, and composite materials were widely used to assist in therapy [4,5]. In recent decades, metallic materials have gained remarkable success due to their excellent mechanical properties [6,7,8]. Stainless steel was the first metal to be used in orthopedics. The addition of chromium, nickel, and molybdenum improved corrosion resistance by forming a tough passive film. Cobalt–chromium alloys have been used in dental applications and recently, in the manufacture of artificial joints [9]. Titanium and its alloys, such as Ti_6_Al_4_V, have been widely used as implant materials in orthopedic surgeries, and have shown excellent performance in electrochemical corrosion properties and a favorable biological response [9,10]. Despite all the advantages of using these materials, there were also some dramatic failures. Placement implants (orthopedic, dental, etc.) can be excellent growth supports for pathogens, which eventually cause the appearance of biofilms. These biofilms can cause major complications once the antibiotic treatments become ineffective due to the difficulty for the antibiotic to reach the biofilm [11,12,13,14,15]. The therapeutic responses currently used are therefore solutions for curative purposes and are generally quite heavy, most often involving a second surgical operation [16,17]. In this context, it is essential to develop preventive rather than curative solutions, to avoid bacterial colonization at the end of the surgical act. The choice of the material and antibacterial agent is crucial to guarantee both an effective action against microorganisms and harmlessness to the human body, and in the best case, favorable biological activity (osteoconduction, osteointegration, etc.) [18,19,20].

It has been reported that the use of bioactive glass can stimulate good functioning of the implant due to its ability to increase tissue integration and enhance its regeneration [21,22,23]. Based on the inorganic composition of natural bone, Hench stipulated that a biomaterial capable of forming hydroxyapatite in an in vivo environment would be able to replace damaged bone tissue without being rejected by the human body [21,22]. Thus, the 45S5 bioglass^®^, composed of 45% SiO_2_, 24.5% Na_2_O, 24.5% CaO, and 6% P_2_O_5_ (wt%), was produced. It represents one of the first examples of a bioactive glass capable of intimately and firmly bonding chemically to surrounding bone tissue without being rejected by the living environment and is considered to be the ancestor of the latest generation of bioactive materials. Indeed, when subjected to an in vivo environment, the bioglass starts to release ions (Na^+^, P^5+^, Ca^2+^), which leads to the formation of a silanol dioxide layer on the surface [24,25]. This layer attracts ions, such as calcium and phosphate, which at a high concentration entails the formation of a phosphocalcic layer on the surface of the glass, similar in composition to the mineral phase of bone [26,27,28]. This apatite layer then allows for the absorption of proteins and the adhesion of cells that proliferate, differentiate, and secrete collagen [29]. The incorporation of collagen fibrils into the growing apatite layer results in a microstructure similar to that of the ligament–bone interface, which explains the important integration of bioglass within host bone tissue [30].

Recently, many efforts have been made to promote angiogenesis, regeneration, and the antibacterial potential of bioglass by the insertion of metal ions in the glass network [31,32,33,34,35,36,37]. Copper is one of the necessary elements for the human body, playing a critical role in angiogenesis and the regeneration of hard and soft tissue [38,39]. Several studies performed on bioactive glass (BG) have shown that the incorporation of Cu significantly enhances angiogenesis by stabilizing the expression of the hypoxia-inducible factor (HIF-1α) in human bone marrow stromal cells (hBMSC) [40,41]. From the bioactivity point of view, the incorporation of copper into the BG network does not provoke any adverse effect, i.e., the formation of hydroxyapatite precipitation on the bioglass surface is preserved after contact with the biological body [40,42]. Beyond being useful in stimulating tissue regeneration, copper is used for its potential antimicrobial effect against several pathogenic bacteria, such as *Staphylococcus aureus*, *Escherichia coli*, *Pseudomonas aeruginosa*, and *Staphylococcus epidermis* [43,44,45,46,47]. All these promising properties make copper a promising ion to be inserted into bioglass to fabricate a multifunctional material for implant coating that combines osteoconduction and osteogenesis with novel therapeutic functionalities.

This work aims to develop 45S5 bioglass^®^ modified by copper oxide insertion to be applied as a coating material for implants. The effect of copper doping on the structure and the morphology of the bioglasses prepared by melt–quenching was investigated in this study. Despite extensive research on bioglasses, few studies have thoroughly explored the relationship between the structure of the 45S5 bioglass^®^ modified with copper oxide and its electrical and bioactive properties. Moreover, the examination of the electrical properties of copper oxide-modified bioglass is notably innovative and has not been explored in prior studies. Changes in the electrical properties were verified due to the ability of these materials to electrically polarize thus optimizing the osseointegration responses. The bioactivity of these glasses was also evaluated in vitro through an immersion test in simulated body fluid (SBF).

## 2. Materials and Methods

### 2.1. Glass Synthesis

Both base and modified bioglasses had been synthesized based on the composition of 45S5 (45% SiO_2_, 24.5% Na_2_O, 24.5% CaO, and 6% P_2_O_5_ (wt%)) proposed by Larry L. Hench [8]. The bioactive glass composition of 45S5 was studied by the introduction of various concentrations of copper, CuO, from 0 to 8 mol% (designed by BG0, BG0.5, …, BG8). In the synthesis of bioglasses, high-purity grade (>99%) SiO_2_, P_2_O_5_, CaCO_3_, Na_2_CO_3_, and CuO, supplied by Sigma-Aldrich, Darmstadt, Germany, were used as the starting compositions. These materials were mixed and homogenized in an agate vessel with milling agate balls for 1 h at 300 rpm, using a high-energy planetary ball milling system [48]. The mixture was then calcinated for 8 h at 800 °C and, afterward, was melted in platinum crucibles that were placed in an electric furnace at 1300 °C for 1 h. The homogeneity was ensured by repeated hand mixing of the melt. Effective cooling was achieved by quenching the molten glasses after removal from the furnace in between casting plates to obtain bulk samples.

### 2.2. Thermal Analysis

Differential Thermal Analysis (DTA) measurements were simultaneously used to examine the thermal characteristics of the glasses. A Hitachi STA 7300 system was used for those measurements, which were performed under Nitrogen N50 (99.999%) flowing at 200 mL/min with a heating rate of 10 °C/min.

### 2.3. Structural and Morphological Characterization

The X-ray diffraction, XRD, patterns were acquired at room temperature using a Malvern Panalytical Aeris powder diffractometer adopting CuKα radiation (λ = 1.54056 Å). The measurement parameters had a scan step of 0.02° in 1 s, in a 2θ angle range of 10–60°.

The Raman spectroscopy of the bulk glasses was performed on a Jobin Yvon HR 800 spectrometer with an Ar^+^ laser (λ = 532 nm), and the spectra were acquired in a back-scattering geometry between 200 and 1500 cm^−1^ using a 50X lens to focus the sample.

The morphologies of the samples were analyzed by TESCAN Vega 3 scanning electron microscopy (SEM). The bulk samples were coated with carbon before microscopic observation. A Bruker EDS system was used in conjunction with the TESCAN Vega 3 microscope to perform a semiquantitative evaluation of the chemical elements on the surface of the samples. The measurements were taken at several surface sites using a 5 µm diameter electron beam spot.

### 2.4. Electrical Characterization

For the electrical analysis, the bulk glass samples were polished to obtain parallel surfaces with a thickness of around 1 mm. Silver conducting paste was applied to the opposite parallel sides of the samples. AC electrical conductivity (σ_ac_) was measured by an Agilent 4294A precision impedance meter, in the C_p_–R_p_ configuration, at the temperature range of between 200 and 400 K with 5 K steps, and in a broad frequency window from 100 Hz to 1 MHz. The dielectric behavior was investigated with the complex permittivity ε* and the complex electric modulus M* formalisms, as expressed by [49,50,51,52]:ε* = ε′ − j ε″ = C_p_ (d/ε_0_ A) − j d/(ω R_p_ ε_0_ A),(1)
M* = 1/ε* = M′+ iM″ = ε′/(ε′^2^+ ε″^2^) + i ε″/(ε′^2^+ ε″^2^),(2)
where C_p_ and R_p_ are the measured capacitance and resistance, d is the sample thickness, A is the electrode area, ω is the angular frequency, and ε_0_ is the permittivity of the free space (8.8542 × 10^−12^ F/m).

The complex AC conductivity (σ_ac_*) was determined using the following relation [53,54]:σ_ac_* = ε_0_ ω ε″+ j ε_0_ ω ε′,(3)

The direct current (DC) conductivity measurements were carried out using a 617 Keithley electrometer. The measurement was performed at the temperature range of between 200 and 400 K, where a DC voltage of 100 V was applied across the bulk glass.

The activation energy (E_A_) for the high temperature range was determined in both AC and DC by fitting the data to the Arrhenius equation [50,53,55]:σ = σ_0_ exp (−E_A_/(k_B_ T)),(4)
where σ_0_ is a pre-exponential factor, E_A_ is the activation energy, k_B_ is the Boltzmann constant, and T is the temperature.

### 2.5. In Vitro Bioactivity Evaluation

The bioactivity test was performed on 7 mm diameter pressed pellets. The assessment of bioactivity was conducted following the “ISO 23317—Implants for surgery—In vitro evaluation for the apatite-forming ability of implant materials” standard [56]. After intervals of 24, 96, and 336 h of immersion in simulated bodily fluid (SBF) with stirring, the samples were withdrawn from the medium and rinsed with deionized water. To create an environment close to the biological one, the medium was changed every 48 h.

## 3. Results and Discussion

### 3.1. Thermal Analysis

The thermal response of the bioglasses is illustrated in Figure 1. The thermogram of BG2 and BG8 shows the existence of a glass transition temperature, T_g_, followed by an exothermic peak, T_c_, attributed to the structural modifications related to the formation of crystalline phases. At higher temperatures, an endothermic peak, T_m_, is assigned to the melting point of bioglass [57,58].

In a previous study [59], a thermal analysis of the 45S5 bioglass^®^ was conducted, revealing a thermal response similar to that of the modified bioglasses. The characteristic temperature values of modified glasses in comparison with the 45S5 bioglass^®^ are shown in Table 1. It can be noted that the characteristic temperatures decrease as the content of CuO introduced into the glass network increases. These results align with those reported in the literature [60]. The changes in the glass temperature might be explained by the type of chemical bonds in the bioglass structure. Due to the stronger affinity of copper to phosphate than to silica groups, the P–O–P bonds were more easily broken compared to Si–O–Si chemical bonds [61]. Thus, Cu–O ionic bonds were created. These bonds have a more covalent character (the ionicity iG of Cu–O bonds is equal to 0.617) and replace the more ionic bonds such as Ca–O (iG = 0.707) [62]. As a result, the thermal resistance of the glasses is reduced, which could explain the decrease in T_g_, T_c_, and T_m_.

### 3.2. Structural Characterization

The XRD patterns of the prepared bioglasses, indicated in Figure 2, show an amorphous hump arising from the glasses having no long-range atomic order in their molecular arrangement. The similarity in the XRD patterns of all the samples demonstrates that the structure of the glass was not affected by the applied exchange conditions.

Figure 3a displays the Raman spectral measurement, which clearly shows that the different bioglasses exhibit a very similar spectrum. However, at a high CuO content, two bands assigned to A_g_ and B_g_ modes of CuO appear at 292 and 568 cm^−1^, respectively [14,63]. A Gaussian fitting was used to deconvolve the Raman spectra of the bioglass base for a more thorough investigation (Figure 3b). In silicate glasses, the vibrational modes at high wavenumbers (>800 cm^−1^) are considered relevant. Six vibrational modes located at around 855 cm^−1^, 903 cm^−1^, 938 cm^−1^, 967 cm^−1^, 1018 cm^−1^, and 1067 cm^−1^ can be observed, which are attributed to the symmetric stretching of Q_0_ Si, Q_1_ Si, Q_2_ Si, Q_0_ P, Q_1_ P and Q_3_ Si units, respectively [64,65,66,67].

Figure 4 depicts the sum of the area of the Raman vibration bands associated with non-bridging oxygen (NBO) ions, i.e., the sum of Q_0_, Q_1_, Q_2_, and Q_3_ units, as a function of CuO content. It can be seen that as compared to the bioglass base, the concentration of NBO ions increases with the increasing CuO concentration of up to 0.5%, then it decreases with further increases in CuO content.

### 3.3. Morphological Characterization

The SEM micrographs, represented in Figure 5, reveal spherical inclusions in the amorphous matrix in both the free and fracture surfaces. The morphology confirms its glassy structure.

### 3.4. Electrical Properties

Figure 6a,b depict the frequency dependence of the dielectric permittivity ε′ and the loss factor ε″, respectively, for the BG2 glass. In this representation, the presence of dielectric relaxation behavior was not observed. In the high temperature and low frequency region, those variations show a linear increase with a slope of ε″ close to −1 (m = −0.95 at 400 K—Figure 5b) thus indicating the existence of the DC conductivity effect [68]. The frequency dependency of AC conductivity may be used to detect this effect. The appearance of a horizontal plateau at low frequencies correlates to the DC conductivity effect, as seen in Figure 6c.

To minimize the electrode polarization and conductivity effects, the electric modulus (M* = 1/ε*) was used [69]. The presence of a dielectric relaxation was observed, whose maximum shifts to higher frequencies with the increasing temperature (Figure 7a). Thus, the relaxation behavior should be associated with the electrical dipole formed between the network modifier and NBO ions. Figure 7b shows a comparison of the normalized imaginary parts of the electric modulus M″/M″_max_ as a function of frequency for the different CuO contents introduced into the bioglass network.

The results presented in Figure 7b reveal that increasing the CuO concentration to 0.25 mol% causes a shift in the peak of the electrical modulus to a higher frequency, implying a reduction in the relaxation time. With a further increase in CuO concentration, the dielectric relaxation peak shifts towards a lower frequency range, indicating an increase in the relaxation time. The increase in the relaxation time with the insertion of more CuO suggests a decrease in the freedom for dipoles in the glass network to orient with the direction of the applied electric field. These findings indicate that the network of the glass containing a concentration of CuO above 0.25 mol% is more “polymerized” [70]. This change in the glass structure is mainly due to a change in NBO ion content as depicted in Figure 4.

Figure 8a,b display the AC and DC conductivity, in logarithmic scale, versus 1000/T, respectively. For all the samples, an increase in temperature is related to the increase in the charge carriers’ mobility, and at the high temperature range, this variation becomes linear. This behavior shows that the conductivity is a thermal-activated process and can be analyzed using the Arrhenius formalism (Equation (4)). Thus, the calculated activation energies for both AC and DC conductivity are presented in Table 2.

The activation energy for DC conductivity is higher compared to AC conductivity. This difference arises from the fact that AC conduction is attributed to ion motion over limited distances, while DC conduction entails motion across longer distances. Consequently, AC conduction involves lower barriers compared to DC conduction and therefore, it requires less energy [54]. The results illustrated in Table 2 show an increase in the AC and DC conductivity for the sample modified with 0.25% CuO compared to the bioglass base, then it decreases with the insertion of a higher concentration of CuO. It is known that conductivity within the bioglass system is mostly correlated with the energy carried by the network modifiers, NaO and CaO, whose mobility increases with the rising amount of NBO ions present in the glass network [58,71]. As depicted in Figure 4, the NBO ion amount increases with the introduction of 0.25% CuO into the bioglass, therefore contributing to elevated AC and DC conductivity. However, as the concentration of CuO is further increased beyond 0.25%, the NBO ion amount decreases, leading to a decrease in the AC and DC conductivities.

### 3.5. In Vitro Bioactivity Evaluation

An in vitro experiment was conducted to evaluate the capacity of the bioactive glasses to facilitate the integration with the host bone and stimulate new bone formation. The test involved observing the development of an apatite layer on the material’s surface after immersion in simulated body fluid (SBF). This technique offers valuable information regarding the physicochemical processes taking place at the interface of the bioactive glass within a biological medium, a crucial factor influencing the adhesion and proliferation of osteoblast cells [72]. It is worth noting that the biocompatibility of these bioglasses was evaluated in our previous work [39]. The SEM micrographs, illustrated in Figure 9, show the surface of the samples after 24 h, 96 h, and 336 h of SBF immersion. It is visible for all samples that there is a formation of spherical particles on the surface, with their size increasing with immersion time. The surface of the pellet becomes fully covered by the precipitated apatite layer and results in a cauliflower shape. The results suggest that the bioglass modified with copper shows promise as an osteoconductive material. Table 3 shows the variation in particle size observed on the surface of the bioglass with immersion time. Compared to the base bioglass, the bioglasses modified with low concentrations of CuO (0.25 and 0.5 mol%) exhibit larger particle sizes even after 24 h of SBF immersion. This suggests that the incorporation of CuO at these concentrations enhances the bioactivity of the bioglass. However, the insertion of a high content of CuO decreases the bioactivity of the glass.

The atomic elements presented on the surface of the prepared glasses were examined using SEM–EDS. The obtained results, illustrated in Table 4, show a decrease in the Si and Na concentrations with increasing immersion time, associated with the dissolution of these elements into the medium. Within the first days of SBF immersion, the Ca/P ratio approaches a value close to the Ca/P ratio of hydroxyapatite in normal bone (Ca/P ≈ 1.67), confirming the formation of the apatite layer [73,74]. The bioglass modified with 0.5 mol% CuO exhibits a Ca/P ratio of 1.71 after 96 h of immersion in SBF, whereas the bioglass base reaches a ratio of 2.05. This suggests that copper oxide has a beneficial impact on the glass’ bioactivity, enhancing the bioactivity within the initial days.

The pH level of the SBF at different times after soaking the bioglass samples is illustrated in Figure 10.

It is noted that from the first few hours, the pH level increases compared to the initial pH of the SBF medium, which was 7.4. This increase in pH continues up to 48 h, then it decreases with increasing immersion time. It is worth noting that the SBF medium was changed every 48 h to simulate the physiological condition. The decrease in pH level can be ascribed to the development of the apatite layer on the bioglasses’ surface [75].

## 4. Conclusions

The present investigation discloses the synthesis of 45S5 bioactive glasses modified by the insertion of CuO using the melt–quenching technique. The structural characterization shows that the glass matrix was not altered by the addition of copper. The deconvolution of the Raman spectra showed an increase in the NBO ion amount with the insertion of CuO. Nevertheless, increasing the concentration of this oxide inserted into the glass network decreases the NBO ion levels. This change in NBO ion amount impacts network modifier mobility, resulting in an increased conductivity for the sample with 0.25% CuO. Bioactivity assessment confirms the glasses’ ability to form an apatite layer on the surface, ensuring a strong connection with bone when applied in regenerative medicine.

## Figures and Tables

**Figure 1 biomimetics-09-00213-f001:**
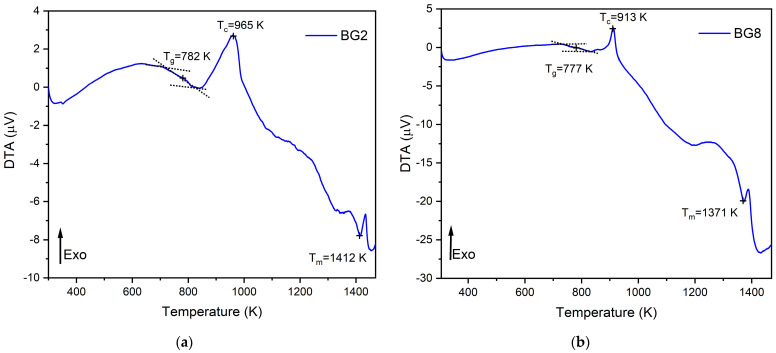
DTA spectra of (**a**) BG2 and (**b**) BG8 samples.

**Figure 2 biomimetics-09-00213-f002:**
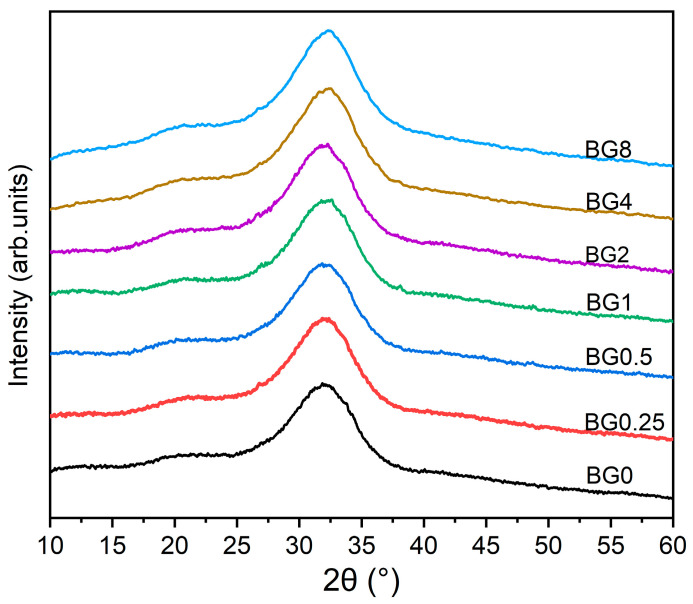
XRD patterns of bioglasses modified with CuO.

**Figure 3 biomimetics-09-00213-f003:**
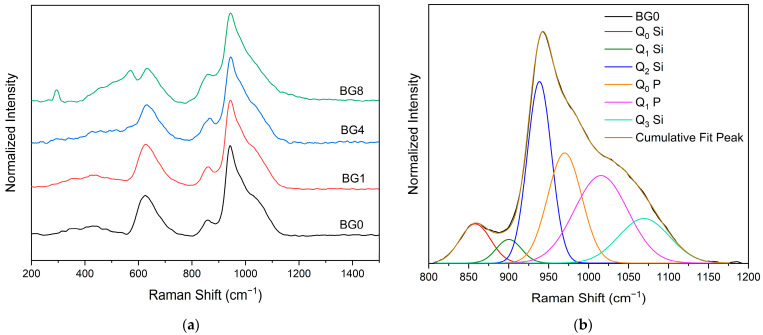
(**a**) Raman spectra of bioglass samples and (**b**) Deconvoluted Raman spectra of the BG0 sample.

**Figure 4 biomimetics-09-00213-f004:**
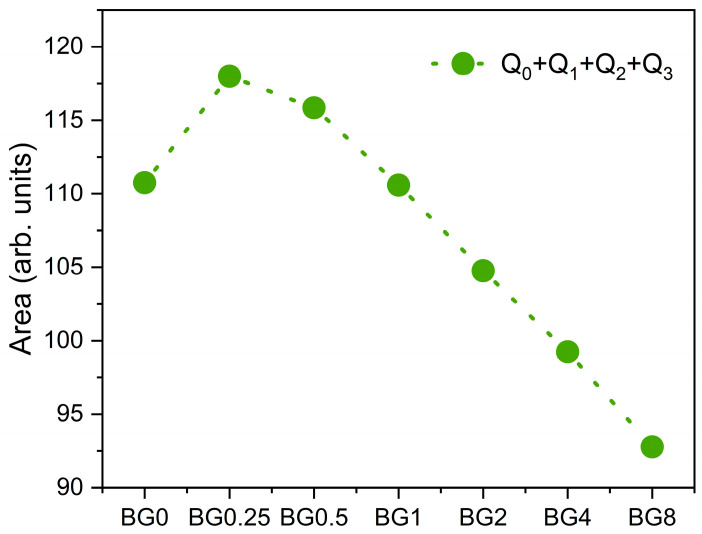
The sum of the areas of the bands associated with NBO ion vibrations.

**Figure 5 biomimetics-09-00213-f005:**
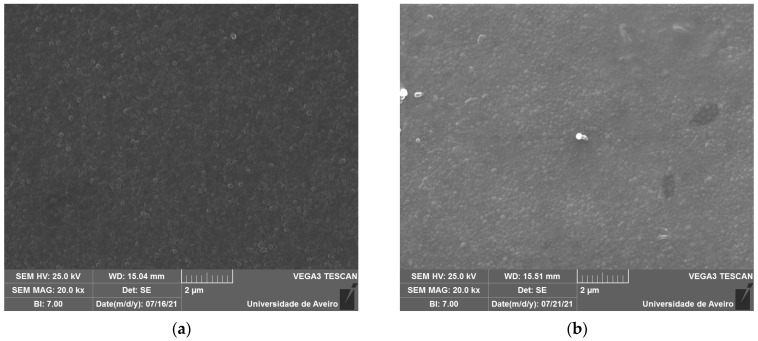
SEM micrograph of (**a**) BG0 and (**b**) BG2.

**Figure 6 biomimetics-09-00213-f006:**
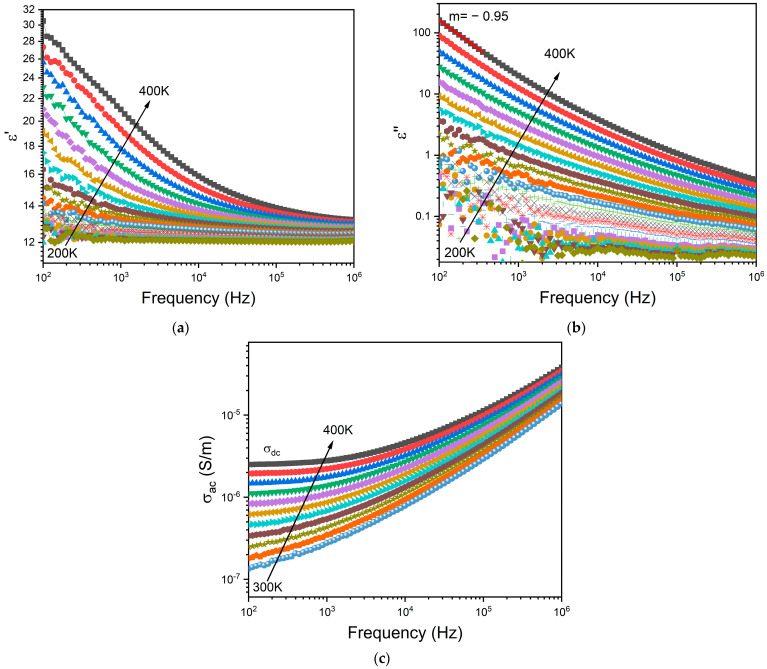
Frequency dependence of (**a**) real part ε′ and (**b**) imaginary part ε″ of the dielectric permittivity, and the (**c**) AC conductivity for BG2.

**Figure 7 biomimetics-09-00213-f007:**
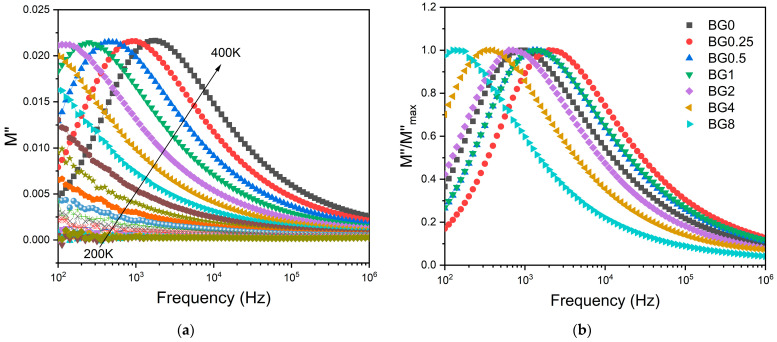
(**a**) The imaginary part of the dielectric modulus M″ versus frequency for BG2 sample; and (**b**) the normalized imaginary part of the modulus M″/M″_max_ versus the frequency at 390 K for all bioglass samples.

**Figure 8 biomimetics-09-00213-f008:**
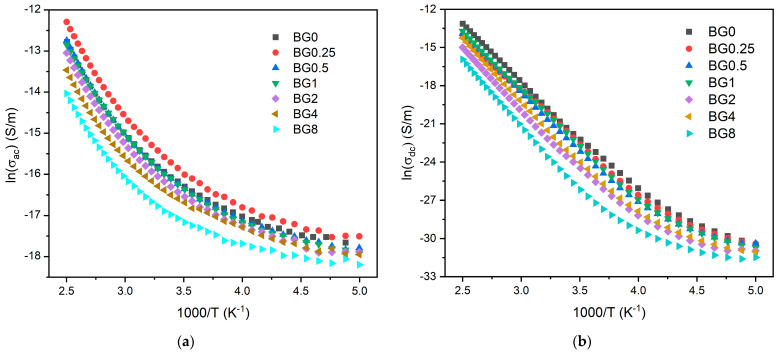
(**a**) AC conductivity versus 1000/T at 10 kHz and (**b**) DC conductivity versus 1000/T.

**Figure 9 biomimetics-09-00213-f009:**
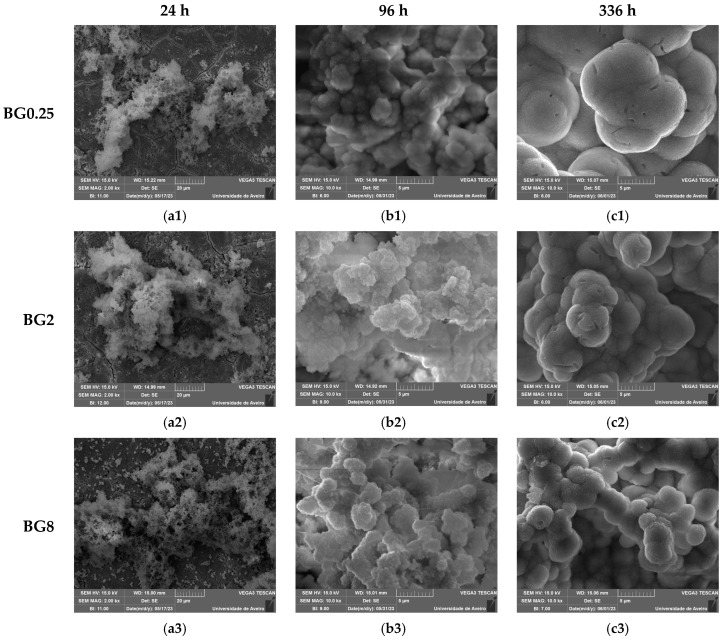
SEM micrographs of the surface of the bioglasses after immersion in SBF for (**a1**–**a3**) 24 h; (**b1**–**b3**) 96 h; (**c1**–**c3**) 336 h. (The magnification of SEM images is 10 kX).

**Figure 10 biomimetics-09-00213-f010:**
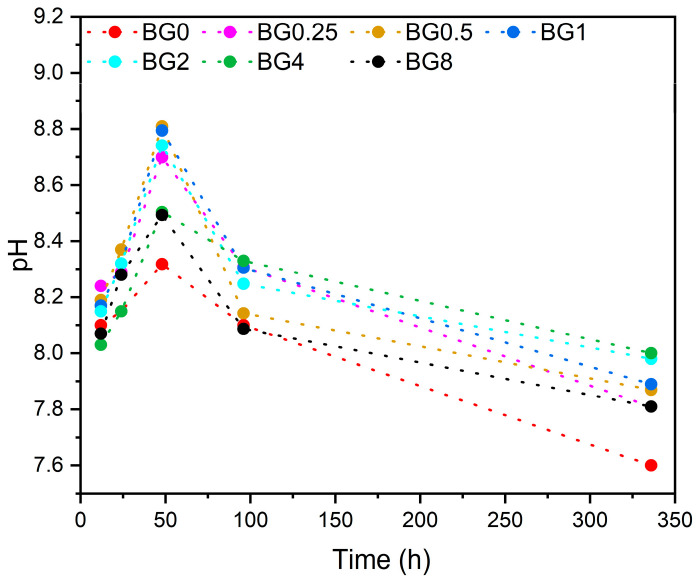
Variation in pH of the SBF medium with the immersion time.

**Table 1 biomimetics-09-00213-t001:** The characteristic temperatures of BG0, BG2, and BG8.

Samples	T_g_ (K)	T_c_ (K)	T_m_ (K)
BG0 [59]	825	1001	1448
BG2	782	965	1412
BG8	777	913	1371

**Table 2 biomimetics-09-00213-t002:** The dielectric constant (ε′), dielectric loss (tan δ), AC conductivity (σ_ac_), AC activation energy Ea (AC), DC conductivity (σ_dc_), and DC activation energy Ea (DC) for all bioglass samples.

Sample	ε’	tan δ (10^−2^)	σ_ac_ (10^−6^) [S/m]	E_a_ (AC) [kJ/mol]	σ_dc_ (10^−9^) [S/m]	E_a_ (DC) [kJ/mol]
	(300 K; 10 kHz)			(10 kHz)	(300 K)	
BG0	13.59 ± 0.72	1.58 ± 0.02	11.92 ± 0.01	37.95 ± 0.98	0.91 ± 0.08	75.82 ± 0.79
BG0.25	13.75 ± 1.42	2.21 ± 0.01	17.12 ± 0.03	38.89 ± 0.73	1.27 ± 0.11	74.27 ± 0.77
BG0.5	11.12 ± 1.24	1.81 ± 0.06	11.37 ± 0.04	38.05 ± 0.79	1.02 ± 0.13	77.24 ± 0.11
BG1	10.14 ± 0.98	2.01 ± 0.03	11.23 ± 0.02	37.37 ± 0.70	1.07 ± 0.15	77.12 ± 0.38
BG2	12.66 ± 1.32	1.32 ± 0.05	9.46 ± 0.08	37.32 ± 0.85	0.23 ± 0.05	83.51 ± 0.12
BG4	12.26 ± 0.83	1.14 ± 0.07	7.66 ± 0.05	37.81 ± 0.86	0.28 ± 0.09	84.00 ± 0.19
BG8	12.34 ± 1.12	0.64 ± 0.02	4.61 ± 0.09	34.53 ± 0.88	0.06 ± 0.001	87.47 ±0.26

**Table 3 biomimetics-09-00213-t003:** The mean sizes of the apatite particles (µm) on the bioglass surface after SBF immersion.

Samples	24 h	96 h	336 h
BG0	0.19 ± 0.06	2.18 ± 0.27	6.58 ± 0.74
BG0.25	0.24 ± 0.07	2.29 ± 0.38	6.63 ± 0.97
BG0.5	0.26 ± 0.05	2.76 ± 0.53	6.97 ± 1.07
BG1	0.18 ± 0.03	2.12 ± 0.29	6.15 ± 0.86
BG2	0.16 ± 0.03	2.22 ± 0.61	4.53 ± 0.52
BG4	0.11 ± 0.04	1.82 ± 0.48	3.98 ± 0.69
BG8	0.05 ± 0.01	1.39 ± 0.31	3.03 ± 0.39

**Table 4 biomimetics-09-00213-t004:** The atomic percentage of Si and Na elements and the Ca/P ratio measured using SEM–EDS, on the surface of the samples before and after immersion in SBF for 96 h and 336 h.

Samples	Si (at. %)			Na (at. %)			Ca/P		
	0 h	96 h	336 h	0 h	96 h	336 h	0 h	96 h	336 h
BG0	11.62 ± 1.1	1.12 ± 0.5	0.11 ± 0.01	15.43 ± 1.1	3.53 ± 0.8	1.31 ± 0.1	7.02 ± 0.9	2.05 ± 0.3	1.78 ± 0.7
BG0.25	11.60 ± 0.9	1.43 ± 0.3	0.52 ± 0.08	15.17 ± 1.3	3.73 ± 0.7	1.27 ± 0.5	5.94 ± 0.7	1.77 ± 0.5	1.71 ± 0.8
BG0.5	11.58 ± 0.7	1.14 ± 0.7	0.14 ± 0.03	15.14 ± 0.9	3.13 ± 0.4	1.26 ± 0.7	5.84 ± 0.7	1.71 ± 0.8	1.70 ± 0.3
BG1	10.23 ± 1.3	1.29 ± 0.1	0.06 ± 0.01	14.91 ± 1.5	4.07 ± 0.3	1.49 ± 0.3	6.66 ± 0.5	1.80 ± 0.7	1.75 ± 0.4
BG2	9.25 ± 0.8	1.47 ± 0.2	0.1 ± 0.04	15.42 ± 1.7	3.76 ± 0.1	1.22 ± 0.2	6.87 ± 0.3	1.74 ± 0.4	1.73 ± 0.5
BG4	10.27 ± 0.6	1.03 ± 0.7	0.08 ± 0.01	13.97 ± 1.2	3.97 ± 0.9	1.18 ± 0.8	6.81 ± 0.9	1.81 ± 0.6	1.78 ± 0.9
BG8	9.34 ± 1.2	1.09 ± 0.3	0.07 ± 0.02	14.84 ± 1.3	4.32 ± 0.2	1.68 ± 0.9	6.71 ± 0.4	1.83 ± 0.5	1.79 ± 0.2

## Data Availability

The data presented in this study are available from the corresponding author upon request.

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
