# Peer review of "Exploring the Impact of Copper Oxide Substitution on Structure, Morphology, Bioactivity, and Electrical Properties of 45S5 Bioglass®"

_biomimetics, 2024, doi:10.3390/biomimetics9040213_

Round 1

Reviewer 1 Report

Comments and Suggestions for Authors

This study aimed to improve osseointegration and reduce implant rejection by developing a biomaterial for implant coatings, utilizing 45S5 Bioglass modified with copper oxide. The modifications enhanced the electrical properties and bioactivity of bioglass, showing potential for improved bone regeneration through the formation of a Ca-P-rich layer on the implant surface. In this respect, the study is novel and the methodology and instrumentation used in this study is satisfying.

Introduction

Line 35-37, 55. Please insert relevant citations at the end of each statement.

Regarding the last paragraph: Please emphasize the novelty of your work and indicate its difference from the literature studies.

Materials and methods section is enough and well designed.

Results and Discussion

The work is very well organized and structured. The presentation and discussion of the results are generally quite good. I have some comments:

Line 158-162, 236-237: Please support your statements with the literature.

Table 2 and Table 3: Apply statistics and make your discussion based on this. Do the values significantly differ or not?

Author Response

Point 1: Introduction

Line 35-37, 55. Please insert relevant citations at the end of each statement.

Regarding the last paragraph: Please emphasize the novelty of your work and indicate its difference from the literature studies.

Response 1:

we inserted in the section “1. Introduction” page 2, line 35-37 and line 55 the citations.

We added in the section “1. Introduction” page 2, lines 93-98 the following paragraph:

 Despite the extensive research on bioglasses, few studies have thoroughly explored the relationship between the structure of the 45S5 bioglass modified with copper oxide and its electrical and bioactive properties. Moreover, the examination of the electrical properties of copper oxide-modified bioglass is notably innovative and has not been explored in prior studies.

Point 2: The work is very well organized and structured. The presentation and discussion of the results are generally quite good. I have some comments:

Line 158-162, 236-237: Please support your statements with the literature.

Table 2 and Table 3: Apply statistics and make your discussion based on this. Do the values significantly differ or not?

Response 2:

we added references to the statement in Lines 158-162, and Lines 236-237.

The statistics and uncertainty analyses were already performed, and the discussion was written based on that.

Reviewer 2 Report

Comments and Suggestions for Authors

This is an interesting paper, and the work is of interest. However, the manuscript is clearly written, but I have some problems with the presentation of the data. This should not be too difficult for the authors to address:

- Did the authors check the ion release as well as the pH of the current modified glass?

- Could the authors point out the difference between the original bioglass and the modified ones, in terms of bioactivity?

Author Response

Point 1: Did the authors check the ion release as well as the pH of the current modified glass?

Response 1:

This study involves the evaluation of the bioactivity of the bioglass samples modified with copper for future application as a coating material for implants. For this purpose, an ISO 23317 standard “Implants for surgery—In vitro evaluation for the apatite-forming ability of implant materials” was used. According to the standard, the bioactivity of the sample is evaluated by examining the formation of a calcium phosphate layer on the surface of the sample. For that, in this study, the SEM-EDS analysis was used to observe the formation of this layer after immersion of the sample in SBF and to evaluate the chemical element on the surface. Moreover, the EDS analysis shows a decrease in Si and Na concentration with increasing immersion time which is attributed to the dissolution of these elements into the surrounding medium.

We added in Section 3 “Results and Discussion” lines 305-313 the following text:

The pH level of the SBF after soaking the bioglass samples at different times is illustrated in Figure 10.

Figure 10. variation of pH of SBF medium with the immersion time.

It is noted that from the first few hours, the pH level increased compared to the initial pH of the SBF medium, which was 7.4. This increase in pH continues up to 48 hours, then it decreases with increasing immersion time. It worth to note that the SBF medium was changed every 48h to simulate the physiological condition. The decrease in pH level can be ascribed to the development of the apatite layer on the bioglasses' surface [75].

Point 2: Could the authors point out the difference between the original bioglass and the modified ones, in terms of bioactivity?

Response 2:

The samples modified with low concentrations exhibited enhanced bioactivity, as evidenced by the rapid decrease in their Ca/P ratios, reaching values close to that of hydroxyapatite (Ca/P ≈ 1.67), compared to the base bioglass. we added in section 3 “Result and Discussion” lines 297-300 the following text:

The bioglass modified with 0.5 mol% CuO exhibits a Ca/P ratio of 1.71 after 96 h of immersion in SBF, whereas the bioglass base reach a ratio of 2.05. This suggests that copper oxide has a beneficial impact on the glass's bioactivity, enhancing the bioactivity within the initial days.

Moreover, We added in the same section  lines 282-287 the following text:

Table 3 shows the variation in particle size observed on the surface of the bioglass with immersion time. Compared to the base bioglass, the bioglasses modified with low concentrations of CuO (0.25 and 0.5 mol%) exhibit larger particle sizes even after 24 h of SBF immersion. This suggests that the incorporation of CuO at these concentrations enhances the bioactivity of the bioglass. however, the insertion of high content of CuO decreases the bioactivity of the glass.

Table 3. The mean size of apatite particles (µm) on the bioglass surface after SBF immersion

Samples

24 h

96 h

336 h

BG0

0.19±0.06

2.18±0.27

6.58±0.74

BG0.25

0.24±0.07

2.29±0.38

6.63±0.97

BG0.5

0.26±0.05

2.76±0.53

6.97±1.07

BG1

0.18±0.03

2.12±0.29

6.15±0.86

BG2

0.16±0.03

2.22±0.61

4.53±0.52

BG4

0.11±0.04

1.82±0.48

3.98±0.69

BG8

0.05±0.01

1.39±0.31

3.03±0.39

Reviewer 3 Report

Comments and Suggestions for Authors

The main work of this study- Exploring the Impact of Copper Oxide Substitution on Structure, Morphology, Bioactivity, and Electrical Properties of 45S5 Bioglass- is to develop melt-quenching 45S5 bioglass modified by copper oxide addition for potential applications as an implant. The manuscript is interesting, but some points need to be addressed before publication.

  1. The concern is that the authors did not show the biocompatibility of the obtained bioglasses.
  2. The in vitro bioactivity test was not fully investigated. (FTIR, XRD, and degradation tests should be included in the manuscript). 

Author Response

Point 1: The concern is that the authors did not show the biocompatibility of the obtained bioglasses.

Response 1:

The biocompatibility of the obtained bioglasses was already evaluated in our previous work (https://doi.org/10.3390/jfb14070369). We added in section 3 “Results and Discussions” line 276-277:

It is worth to noting that the biocompatibity of these bioglasses was evaluated in our previous work [40].

Point 2: The in vitro bioactivity test was not fully investigated. (FTIR, XRD, and degradation tests should be included in the manuscript).

Response 2:

We appreciate the review insights. The main objective of our work is to evaluate the relation between the structural characteristics of the prepared bioglasses with their electrical properties since they exhibit the ability to store electrical charge on the surface which could enhance the osseointegration reaction. The evaluation of the bioactivity in this work has like objectify to prove the possibility of these glasses being applied for bone regeneration medicine. In previous studies (https://doi.org/10.3390/jfb14070369/https://doi.org/10.3390/nano13192717/https://doi.org/10.3390/magnetochemistry9090209), the evaluation of the in vitro bioactivity of the bioglasses was performed using the ISO 23317 standard “Implants for surgery—In vitro evaluation for the apatite-forming ability of implant materials”. According to the standard, the bioactivity of the sample can be evaluated by examining the formation of a calcium phosphate layer on the sample's surface using SEM-EDS analysis to observe the formation of this layer after immersion of the sample in SBF and to evaluate the chemical element on the surface. These techniques successfully show the formation of a layer of calcium phosphate covering the whole surface of the sample. We recognize the significance of FTIR for observing changes in chemical structure, XRD– particularly grazing incident X-ray diffraction (GIXRD) – for detecting the crystalline phases of hydroxyapatite formed upon interaction with physiological fluids, and degradation analysis using Inductively Coupled Plasma (ICP) to evaluate ion concentration in the SBF solution over time. These analyses will be the focus of our future work.

Round 2

Reviewer 2 Report

Comments and Suggestions for Authors

The authors have successfully addressed all comments.

Author Response

The authors would like to thank you for your positive feedback regarding our revisions.

Reviewer 3 Report

Comments and Suggestions for Authors

Accept in its present form

Author Response

(The authors gave the same response as above.)
